# Interoperable Blockchains for Highly-Integrated Supply Chains in Collaborative Manufacturing

**DOI:** 10.3390/s21154955

**Published:** 2021-07-21

**Authors:** Paolo Bellavista, Christian Esposito, Luca Foschini, Carlo Giannelli, Nicola Mazzocca, Rebecca Montanari

**Affiliations:** 1Department of Computer Science and Engineering, University of Bologna, 40100 Bologna, Italy; paolo.bellavista@unibo.it (P.B.); rebecca.montanari@unibo.it (R.M.); 2Department of Computer Science, University of Salerno, 84084 Fisciano, Italy; esposito@unisa.it; 3Department of Mathematics and Computer Science, University of Ferrara, 44122 Ferrara, Italy; carlo.giannelli@unife.it; 4Department of Electrical Engineering and Information Technology, University of Napoli “Federico II", 80125 Napoli, Italy; nicola.mazzocca@unina.it

**Keywords:** Industry 4.0, decentralized ledger, interoperability, blockchain, Trusted Execution Environment

## Abstract

Blockchain technology plays a pivotal role in the undergoing fourth industrial revolution or Industry 4.0. It is considered a tremendous boost to company digitalization; thus, considerable investments in blockchain are being made. However, there is no single blockchain technology, but various solutions exist, and they cannot interoperate with one each other. The ecosystem envisioned by the Industry 4.0 does not have centralized management or leading organization, so a single blockchain solution cannot be imposed. The various organizations hold their own blockchains, which must interoperate seamlessly. Despite some solutions for blockchain interoperability being proposed, the problem is still open. This paper aims to devise a secure solution for blockchain interoperability. The proposed approach consists of a relay scheme based on Trusted Execution Environment to provide higher security guarantees than the current literature. In particular, the proposed solution adopts an off-chain secure computation element invoked by a smart contract on a blockchain to securely communicate with its peered counterpart. A prototype has been implemented and used for the performance assessment, e.g., to measure the latency increase due to cross-blockchain interactions. The achieved and reported experimental results show that the proposed security solution introduces an additional latency that is entirely tolerable for transactions. At the same time, the usage of the Trusted Execution Environment imposes a negligible overhead.

## 1. Introduction

Industry 4.0 is a significant technological trend at the crossroads of Industrial Internet of Things (IIoT), Cyber-Physical Systems (CPSs), and digital manufacturing that is gaining increasingly more momentum pushed by big companies and central/regional governments. The promise of Industry 4.0 is manifold and includes delivering new services, enabling new business opportunities, and creating new employment [1,2,3]. Moreover, Industry 4.0 puts forward as a core idea a unified and fully integrated view of manufacturing enterprises that is not only vertical, from the shop floor to the top-level management, but also horizontal, across the whole supply chain. In brief, Industry 4.0 standardization efforts are paving the way to the possibility to support deeply integrated supply chains [2,4,5].

However, some sectors, such as Small and Medium Enterprises (SMEs) in manufacturing, although already adopting very advanced automation solutions on the shop floor several decades ago, are still not ready for the fully-interconnected Industry 4.0 vision. Indeed, the main obstacle is often a rigid separation between the various production departments devoted to performing manufacturing processes (hosting work machines and production lines) and IT departments more committed to managerial tasks. This is also motivated by both security and safety concerns for the shop floor, where software errors may result not only in money but also in life losses. Moreover, the high heterogeneity of IIoT and CPSs complicates trust and security support as devices span from powerful supervision systems to resource-constrained wireless micro-controllers, edge/fog devices, and robotic arms. In addition, Industry 4.0 processes can span the whole vertical spectrum of the enterprise, including both human actions/interventions as well as more technical aspects such as configurations, network and service management, and so forth.

Blockchain technologies can play the role of the integration glue to smoothly bring together all the pieces of this complex human and technical manufacturing ecosystem, by being able to secure it at the same time. In fact, thanks to its inherent ability to enable secure and non-repudiable logging of events in a fully distributed fashion without any centralized authority, blockchain is being increasingly more applied in the industrial manufacturing application domain. That is especially true for those scenarios that require supporting supply chain management. Just to cite a few efforts along that direction, the authors of [6] leverage the Ethereum blockchain for the soybean supply chain, thus removing any central authority by supporting at the same time tracing, tracking, and business transactions. The work in [7] adopts the blockchain in the food sector by leveraging novel solutions to increase food safety, e.g., to manage the supply chain so as to avoid inefficiency, opacity, and fraud as well as to increase the reputation of firms and their food products. Finally, the work in [8] employs a blockchain solution to avoid fraud by using unique identifiers provided by NFC tags.

Nowadays, existing solutions for Industry 4.0 do not include a single infrastructure owned by a single company. Still, instead, they typically consist of an ecosystem of various collaborating companies, each focusing on a specific set of tasks and an established dynamic commercial relationship with the other participants in the ecosystem. Within such a context, it is impossible to impose a single technology and solution to the various companies, so it is more feasible to keep each participant having their own data exchange and management solution and let these solutions interoperate among themselves. Unfortunately, shifting towards blockchain technology does not simplify such a scenario, as the various existing blockchain platforms diverge on the adopted block structure, consensus algorithm, and coding of smart contracts. Such a strong degree of heterogeneity is an obstacle to the seamless interaction among these platforms and, thus, to transferring assets from one blockchain to another. In recent years, some solutions have been proposed to allow blockchain interoperability by adopting centralized mediators, such as the notary schemes, or decentralized sidechains and relays with non-negligible overhead, and hash locking schemes focused only on the cryptocurrency exchange. Despite such solutions, blockchain interoperability is still an open issue, primarily because the security guarantees in these approaches have not been fully investigated and approached.

This work proposes a novel solution based on relay schemes, where a software component forwards possible blockchain updates to the other interconnected ones. Our solution does not need any sidechains, which are chains of blocks holding data coming from the other interconnected blockchain. Instead, it has an off-chain computation element, which is software bringing computations outside of the blockchain and is not subject to its distributed agreement protocol. Such an off-chain relaying element is the one that is contacted by a smart contract to realize cross-chain interactions. Such an element runs on one node of the cooperating blockchains and can securely communicate with its peered counterpart. To secure the overall interoperability operations, such off-chain computation is hosted in the Trusted Execution Environments (TEE), which is a secured execution environment guaranteeing that any code and data internally stored and executed is protected against any possible external, malicious, and non-authorized manumission. Even the user authentication in case of a permissionless blockchain has been implemented within the TEE, to avoid incurring the overhead of having a certification authority, to centrally handle the issue of releasing and managing security and identity claims. Such a solution has been implemented within the context of the Hyperledger Fabric [9] and Sawtooth [10], which are two different solutions despite being included in the Hyperledger family. As a TEE solution, we have used the Open Enclave SDK [11], and openSSL [12] has been exploited as the library offering primitives for secure communication. An empirical assessment has been performed to measure the performance overhead incurred by using the proposed solution concerning the case where operations occur within the same blockchain. TEE has been applied within the blockchain to avoid recurring to Byzantine Fault Tolerant consensus, needed to cope with any possible malicious behavior of the participating nodes, which takes a long agreement time. The most efficient Crash Fault-Tolerant consensus has been proferred, as in a version of the Proof of Elapsed Time (PoET) consensus in Hyperledger Sawtooth [13]. TEE has also been used to protect off-chain computations when executing smart contracts as in Hyperledger Avalon [14]. The main novelty of our approach is to use TEE to secure interoperability operations, which is a problem that has been slightly considered in the current literature.

The paper is structured as follows. Section 2 describes the needed background on blockchain and its interoperability requirements and presents the existing literature on allowing blockchain interoperability. Section 3 illustrates the proposed approach by introducing the concept of inter-blockchain smart contract and specifying the design of a secure TEE-based blockchain interaction scheme. Section 4 has a twofold contribution. On the one hand, it discusses how the proposed approach has been implemented by introducing used building blocks (i.e., Hyperledger Fabric, Sawtooth, and Open Enclave SDK) and how they have been used to realize the interoperability solution. On the other hand, it also shows achieved experimental results based on the implemented proof-of-concept. Last, Section 5 concludes this paper with some lessons learned and a plan for future works.

## 2. Blockchain for Collaborative Manufacturing: State-of-the-Art and Motivation

This section presents needed background material about blockchain in general and, more specifically, blockchain in Industry 4.0. Then, we conclude the section with a in-depth analysis of related works in blockchain interoperability.

### 2.1. Blockchain Background

A blockchain [15] consists of a decentralized technology with a consistent and immutable replication of data or blocks containing details of the transactions undergoing between peers over the Internet, thanks to the integration of a suitable distributed consensus algorithm able to cope with crashes and byzantine failures (i.e., intentional or unintentional and unpredictable deviations from the protocol defining the normal, valid, and correct behavior of the participants to the blockchain). Such technology has not been standardized, but under the umbrella of this term, we can find many different architectures and implementations, each adopting a specific set of design strategies and realizations. All those solutions can be groups in a taxonomy by considering various features and characteristics. A first possible classification can be made by considering if participants to the blockchain are pre-defined and regulated by proper authentication and authorization means or not. On the one hand, there are blockchain solutions, such as Bitcoin or Ethereum, where nodes freely participate with the consensus algorithm and the block replication, and for this reason, these solutions are known as permissionless. On the other hand, other blockchain solutions, such as Hyperledger Fabric, allow nodes to participate in certain functionalities of the solution based on pre-define security policies or by exhibiting proper security claims, and for this reason, these solutions are known as permissioned.

The permissionless or permissioned nature of a blockchain solution implies derived features and characteristics thanks to a proper application of the famous CAP theorem [16]. This theorem states that a distributed system on the Internet can provide only two properties among Consistency, Availability, and network Partition tolerance, while it cannot be guaranteed the reaming property. Within the Internet, tolerating possible network partitions, which can be persistent or temporary and are quite frequent due to network misbehavior and/or faults, is a must and cannot be weakened, so it is always supported. Therefore, a solution needs to determine if sacrificing consistency or availability. By considering permissionless platforms, the distributed consensus being applied is the one designed by Nakamoto or its possible variations where the Proof-of-Work is adopted to make it non-profitable to have behavioral deviation and to encourage honest behaviour [17]. Nakamoto-style consensus allows little time windows where consensus is not achieved, but the blockchain presents forks that are resolved later by having the more extended branch to dominate the other one [18]. Thus, forks are clear evidence that consistency is not always guaranteed, but the system is always responsive (even if inconsistent), implying high availability. Therefore, this indicates that permissionless solutions are clear instances of AP systems. On the contrary, the current permissioned solutions integrate more robust consensus algorithms, derived from the State Machine Replication (SMR) and Byzantine Fault Tolerant (BFT) consensus [17,18], to avoid possible forks and ensure strong consistency among the block views stored by all the participants to the blockchain. While reaching consensus, the system cannot welcome new requests, meaning that permissioned solutions are instances of CP systems.

Another main difference is related to the programming language and the consequent Turing completeness used to realize smart contracts, i.e., distributed applications running on top of the blockchain making the “Transposition” of a contract into code, using “if/then” functions incorporated in software or IT protocols, to verify the fulfillment of certain conditions automatically and to self-execute actions when the conditions are met and verified. The scripting language used in Bitcoin is not Turing complete, as loops are missing. In contrast, the Solidity language used in Ethereum is complete but purposely designed for realizing smart contracts. Last, Hyperledger Fabric contracts use general-purpose languages such as Go, node.js, and Java. Such heterogeneity in the used programming language causes the impossibility of having blockchain platforms cooperating and a smart contract being portable across the various platforms.

### 2.2. Blockchain in Industry 4.0

In this period of industrial ferment, companies need to reflect on digital technologies’ opportunities and redefine the drivers of value. Efficiency and operational effectiveness will receive impetus from the affirmation of Smart Factories and Smart Supply Chains. Moreover, Industry 4.0 will lead to growth opportunities through innovative technological solutions to increase the value for customers. In essence, the new paradigm will take advantage of disruptive technologies, such as IoT and CPSs, which will lead to autonomous communication between many industrial devices located within a factory, thus supporting the remote monitoring and real-time control of machinery.

In this framework, the advent of blockchain technology represents an excellent opportunity to accelerate the digital transformation of industrial processes and to leverage the technologies of Industry 4.0 in different contexts. The modern factory uses networks to connect smart manufacturing systems vertically. By enabling these connections, it is possible to automatically collect and send the information collected by the different systems present within the plant to the various parts of the value chain (be it a project team or line operators). The blockchain can make standard trusted data or money exchange points available through which the different entities of the smart factory can interact. The advantage of counting on the characteristics of a blockchain is evident: decentralization, immutability, and transaction timestamping can be easily supported. All this allows increasing the reliability and trustworthiness of the origin of a specific item, to better track the logistics, to accelerate the acknowledgment of the compliance with the standards, as well as to record any information of interest, such as the operating conditions of a particular system to aim for the best preventive maintenance plan.

Industry 4.0 technologies must also be integrated horizontally: producers, suppliers, and customers must collaborate. In this regard, the blockchain has recently been used as a data exchange backbone solution: the actors involved can enter transactions and trigger events to drive the IoT devices [19]. For instance, by considering industrial equipment management on the shop floor, blockchain adoption allows tracing when and how each tool has been reconfigured in an immutable manner, also identifying the person (or the service) issuing the reconfiguration command in a not repudiable way.

In addition to what has been said, the blockchain represents a useful technology for securely storing information relating to intangible procedures, e.g., when a checkup of an industrial tool was conducted simultaneously with operational data such as peak measured vibrations. Therefore, a key objective for Industry 4.0 smart factories is to allow rapid reactions to the feedback received from the various members of the value chain. Another aspect of Industry 4.0 concerns the integration of different technologies. In this regard, the blockchain can become an information exchange hub. The various users, independent of the technology used, only need to implement the most appropriate blockchain client functionality. Figure 1 schematically illustrates the application of blockchain in the industry. A given industrial process is split among various firms, each devoted to a specific phase and offering the resources for composing the manufacturing process. The recent view of cloud manufacturing [20] consists in integrating all users and all workflows, avoiding waste and unproductive losses along the entire value chain, multiplying the possibilities of negotiating and managing contracts. Specifically, it realizes the outsourcing of systems, infrastructures, and services whose IT components (system hosting, maintenance, updates, and security supervision) instead of being managed directly by the company are entrusted to a trusted provider that provides the service in on-demand and pay-per-use. Nowadays, manufacturing companies are typically large, with factories and warehouses distributed in different geographic locations with national and international supply and subcontracting relationships. Networked manufacturing uses middleware for the integration of these distributed resources. The main limitation of such an architecture is to have the overall production process segmented in portions that communicate among each other with the middleware but lacks a way to provide a global view of the overall process. With cloud manufacturing, it is possible to follow the entire production process, from the design phase to the realization of the product, up to its maintenance. However, such a vision implies a centralized (in the cloud) control of processes, despite nowadays in manufacturing witnessing ecosystems of companies rather than a single big player with many subsidiaries. Blockchains integrated with cloud manufacturing can cope with this limit as presented in [21]: each firm uses a node of a blockchain solution to get data and send commands towards the interconnected production resources, and up to the cloud-hosted management applications that the administrators of the firm use. Such an example makes it clear that the shipping facility needs data from the production facility to plan the logistics. In contrast, the design center needs to send the details to produce products to the production site, and so on.

The current academic literature and industrial practices encompass some recent attempts of applying blockchain within the context of the Industry 4.0 environments, as depicted in Figure 1, not only to support Cloud Manufacturing, but also for IoT enabled manufacturing and service-oriented manufacturing, such as in [22,23,24]. Recent contributions started to focus on Industry 4.0 scenarios [25,26]. For instance, the work in [27] analyzes the benefits of adopting the blockchain in the automotive industry to support trusted and cyber-resilient information distribution among currently non-collaborative organizations. The work in [28] presents a solution adopting the blockchain to support the servitization of ice cream machines by exploiting smart contracts to ensure data validity related to machine usage. Similarly, in [29] the Blockchain is used to allow machine owners to share idle machines capacity by securely storing in the blockchain relevant events related to machine usage. The work in [30] supports actors of the manufacturing supply chain to make agreements and payments based on the blockchain in a secure and distributed manner, without an intermediary. As agreements are stored in transactions and thus impose the payment of a fee, the proposed solution adopts a hybrid approach by allowing actors to pay only for agreements that actually need to be secured. The work in [31] exploits the blockchain to ensure data privacy of IoT devices by adopting smart contracts to validate connection rights based on predefined privacy permission settings predefined and on the availability for each IoT device of a set of stored known misbehaviors. The BPIIoT solution [32] exploits the blockchain to develop an IIoT platform, with the notable benefit of addressing issues related to the lack of security, trust, and island connectivity typical of many IIoT environments. In particular, BPIIoT exploits Blockchain smart contracts as a mechanism to achieve an agreement among service consumers and manufacturing resources supporting the delivery of on-demand manufacturing services. The BASA solution [33] allows cross-domain authentication IIoT environments. In particular, it will enable authenticating devices by other devices, even if in a different administrative domain, also without requiring to expose identity information. In this manner, BASA can ensure trust in untrusted domains without adding any third-party entity.

These attempts have proved to bring several advantages to these environments, but the authors also highlighted some weaknesses to be properly tackled by future research initiatives. Without the intent of being exhaustive, we think that the majority of these problems are as follows:The Industry 4.0 environments are characterized by a significant amount of traffic generated by the applications and devices integrated into the production lines, but blockchain solutions have known scalability limitations due to the adopted consensus algorithms.The consensus algorithm determines the provided QoS of the blockchain, and the user is forced to choose between consistency or availability. The selection of a consensus algorithm in place of another may impact the blockchain latency and throughput and has an impact in the case of running blockchain on resource-constraint devices. In Industry 4.0 environments, it is pivotal to make such a selection by optimizing the trade-off among opposing objectives.Industry 4.0 environments are ecosystems of companies, and it is not sure that all of them may agree on using the same blockchain solution. Still, it is more reasonable to have various blockchain solutions coexisting and interoperating in a single ecosystem. For this aim, it is important to investigate possible interoperability and standardization means.

As analyzed in the following subsection, the available related works in the current research are more focused on the first point by improving two aspects of scalability and latency of the blockchain, while the other matters are scarcely investigated. Mainly, some industrial solutions claim to provide blockchain interoperability. Still, in practice, they support few solutions to be integrated, or only certain kinds of applications (such as transfer or currency or digitalized resources) are supported.

### 2.3. Blockchain Interoperability Related Works

Blockchain technology has been implemented in multiple manners, fostering the spread of heterogeneous platforms and solutions that do not interoperate with each other. While it is generally recognized that blockchain interoperability is strongly needed, the concept of blockchain interoperability is not well defined in the current literature. As an actual example, in [34] two blockchain solutions are assumed interoperable if there exists a transaction in a block accepted by the consensus algorithms running at both solutions and considering their states. However, the authors have proved that such a definition is not feasible and introduced a weaker definition where the consensus algorithms consider the joint state of both blockchains. In [35], blockchain interoperability implies secure state transitions across different blockchains by running cross-chain decentralized applications. The work in [36] proposes a different definition where interoperability is achieved when native cryptocurrencies can be exchanged among two or more heterogeneous blockchain platforms. In [37], blockchains can interoperate if they can mutually exchange information among them in a unified manner. Last, in [38], interoperability is identified not only between smart contracts or data exchange between heterogeneous solutions (i.e., inter-blockchain interoperability) but also within the same distributed ledger where smart contracts designed by different designers and organizations can interact with each other (i.e., intra-blockchain interoperability). Definitely, interoperability does not mean that a given smart contract formalism and programming language used in one platform can also be executed in another one. Therefore, as an example, the deployment of Ethereum Smart contracts written in Solidity on Hyperledger Fabric through the EVM chaincode (EVMCC) and Fab3 [39] is not an example of blockchain interoperability.

Within the context of this work, we envision interoperability as the applicability of smart contacts between two different chains within a single distributed ledger or across heterogeneous solutions. This means that it is needed to securely change the state of the two involved chains as the effect of smart contract execution. We differ from previously mentioned studies as we aim to consistently perform a state change at two chains and prevent possible inconsistencies due to attacks and byzantine behaviors of the involving entities. The main existing solutions have investigated the possibility of transferring an asset (such as an amount of cryptocurrency or data within the chain) from one blockchain platform to another. However, the actual quality of such interoperability solutions has not been thoroughly tested in practice. In addition, the security of the cross-blockchain asset transfer has not been fully considered yet. This makes the existing interoperability solutions vulnerable to many kinds of attacks and failures.

The current literature of the interoperability solutions is not only limited by the inter-ledger communication protocols [40] to allow the communication from one blockchain to another but also includes software products to satisfy some of the previous definitions. Therefore, they can be grouped within three types of architectures [36,38]. The first type employs a centralized approach by having *Notary schemes*, as shown in Figure 2a. Specifically, these solutions have a central trusted component that mediates between interacting blockchain platforms and controls that the asset fetched from one blockchain is correctly added to the state of the other blockchain. Such a design is simple and verifies the consistent asset transfer despite possible failures and attacks. An example of this kind of solution is represented by Hyperledger Cactus [41], which is designed around a routing API and business logic interface to route requests across multiple blockchains and validator nodes to offer proofs of the state changes due to cross-chain transactions.

Such a centralized approach is in opposition to the key rationale behind all blockchain platforms of decentralized control. Therefore, the second type of solution has been designed by implementing *Sidechains* or *relay schemes*, depicted in Figure 2b. Specifically, these solutions are equipped with a given smart contract on the first ledger to access assets or states held by the second blockchain and execute computations. This is possible because a summary of the data within the second blockchain is held in the first one, and such a summary is called sidechain. Such a design is completely decentralized and does not require a central trusted mediator. Still, pegged sidechains are used to lock an asset in one blockchain to reserve it until the transfer to the other one is completed. It is possible to have one-way relays as the interaction is only possible from one blockchain to another, but not vice versa, or two-way relays when the communication between interoperating blockchain is bidirectional. Examples of this second type are Polkadot [42], Cosmos [43], ChainLink [44], or Interledger [45].

Such solutions impose high execution costs due to their need to transfer data from the origin blockchain to a sidechain and back from the sidechain to the intended destination blockchain. Moreover, they face difficulties providing atomic cross-chain swap for two parties, each owning crypto-tokens in different blockchains. The last possible type (presented in Figure 2c) consists of using *hash-locking schemes* for a decentralized and consistent atomic asset transfer among blockchains. Mainly, atomic state changes are simultaneously commenced at heterogeneous platforms, where both changes are either committed or rolled back. The advantage concerning the second type is that these schemes require significantly less information exchange to achieve interoperability. However, these solutions have not found prolific applications as they are only limited to atomic swap, which can still be realized by using the other two types and cannot transfer tokens from one blockchain to another, which are possible with notary and sidechain schemes. Therefore, a cross-chain hash lock is adopted by the Lightning Network [46], which consists of a “layer 2” payment protocol built on top of a blockchain-based cryptocurrency (like Bitcoin) to provide a decentralized network instant payment method across a network of participants. Most of the available solutions belong to the second type, as they are the most simple to implement without breaking the decentralized nature of the blockchains. However, they demand a change to the interconnected blockchains, as proper locking mechanisms must be introduced for consistent state management for cross-chain transactions. Moreover, the asset transfer presents non-determinism within the smart contract by fetching the asset from one blockchain and passing it to the other. Therefore, after the completion of its execution, the state change depends on an external component, i.e., the receiving blockchain.

As Table 1 summarizes, none of the available solutions deals with possible issues caused by failures or attacks occurring during the inter-blockchain asset transfer, with the only exception being that ChainLink uses TEE to protect the execution of the oracles. The trust model typically assumes that the notary node or the connector with the side chain is trusted and not compromised (by exposing a byzantine behavior). In the case of ChainLink, the only application being supported is a transfer of data by transforming (in the TEE-assisted secure environment) data from its incoming links to the expected format/model of the sources behind its outgoing links, without the notion of shared state among blockchains which is, on the contrary, supported by the other solutions. To fill this gap, there is the ongoing integration of ChainLink and Polkadot, leading to the possibility of hosting ChainLink oracles in the framework blockchain Substrate [47] used by Polkadot. However, such integration is not fully complete as the oracle cannot be used within the smart contracts running on Substrate, making the project not mature enough for real usage in real-world scenarios. Thus, the research problem addressed in this work is as follows: How can we protect cross-blockchain transactions when nodes can deviate from the designed protocol and exhibit a Byzantine behavior in cross-blockchain state update? What are the costs, in terms of increased latency, caused by introducing the protection means? In fact, despite the availability of these solutions and some actual examples, it is not possible to get a glimpse of their latency cost.

## 3. Interoperable Blockchains for Collaborative Manufacturing

This section introduces the main design guidelines and the distributed architectural model of our proposal by thoroughly discussing the pros and cons of different interoperability approaches to motivate the employed TEE-based approach.

### 3.1. Design Highlights

The manufacturing supply chain is characterized by several actors interacting with one another in an articulated manner. For instance, a manufacturing company producing industrial equipment sells its tools to other manufacturing companies, crafting final products and providing assistance services. However, the same manufacturing company also receives spare parts and assistance services from other companies of which it is a client. Such a scenario requires companies to exchange information in a well-defined manner easily, i.e., there is the need for allowing blockchains of different companies to generate transactions in a coordinated way.

To better present this aspect, let us consider what happens when a technician of a company providing maintenance services (employed at the company S) verifies and certifies that a tool of a manufacturing company (company M) works properly (see Figure 3). The blockchain of S and M must interact as there is the need, first of all, of verifying that company S is an authorized maintainer of company M with an active agreement (information A on blockchain M) and that the technician of S owns required certifications to manage company M industrial equipment (information B on blockchain S). Then, once requirements have been verified and the maintenance performed, the smart contract must be able to trigger the generation of new data in both company M and company S blockchains, specifying the positive/negative outcome about the maintenance procedure with related details (transaction X in blockchain M) and the accomplishment of the required task performed by the technician of company S (transaction Y in blockchain S).

As is better detailed in the following, such a scenario can be enabled by the federation of blockchains based on the enforcement of inter-blockchain smart contracts. Each blockchain is managed by a single company in a manner fitting its own requirements. Thus, each blockchain contains (private) smart contracts and enforces transaction creation policies specifically identified by considering the capabilities and constraints of the company’s processes. Moreover, different companies interacting with one another, e.g., based on maintenance agreements, should develop and deploy proper smart contracts allowing the generation of transactions in different blockchains but in a coordinated manner, based on constraints ruling inter-company interactions. The purpose of inter-blockchain smart contracts is to verify the fulfillment of each blockchain’s requirements in a coordinated manner. Only in the case of a positive response should the inter-blockchain smart contract actually trigger new transactions in the two blockchains.

### 3.2. Distributed Architecture and Interoperable Blockchain

The inter-blockchain smart contract approach briefly sketched in Figure 3 has the primary characteristic of not requiring the deployment of a centralized solution based on a notary scheme. Instead, its approach is to split the logic of interoperable smart contracts into two parts, one interacting with a blockchain and the other one with another interoperable blockchain, like what happens in the case of two-way relays. Moreover, we have discarded the possibility of using a sidechain, with the primary goal of avoiding memory occupation and time consumption to keep such a chain consistent with the main chain. Furthermore, we have assessed hash locking schemes as not being suitable due to their limited applicability in the target scenario [38]. Hash locks are mechanisms applied to cryptocurrencies to reserve transactions on one ledger until the second ledger produces the cryptographic proof based on the correct execution of an atomic swap. This is needed so that both ledgers have a consistent state due to the inter-blockchain smart contract execution. However, current solutions for hash locking have not been used outside the context of cryptocurrencies, such as token portability or cross-chain oracles. Specifically, the physical objects are represented within a blockchain utilizing tokens whose format is platform-specific and hard to exchange among platforms due to the representation heterogeneity. A cross-chain oracle is a smart contract running at one ledger, reading from a smart contract on another ledger, and performing an action based on the received read result.

Figure 4 takes inspiration from, and relevantly extends, the solution sketched in Figure 3. In particular, Figure 4 outlines the proposed inter-blockchain smart contract implementation approach, where an inter-blockchain smart contract designed to transfer assets (such as certain amounts of cryptocurrency or tokens) is split into two separate contracts and deployed in two different ledgers (left and right). The user invokes only the smart contracts fetching the asset (the left one in the figure) that are in charge of getting the current status of the hosting ledger and of interacting with a second counterpart in the other interoperating ledger (the right one). The contacted smart contract on the right gets the state of its hosting ledger and, based on the retrieved information, decides if and how to update the ledger with the new asset. Before terminating, the smart contract on the right informs the other smart contract about the operation’s completion so that the latter can appropriately update the state on its blockchain. Note that the attack surface of this approach consists of eventual network misbehaviors, e.g., discarding or reapplying exchanged messages or behavioral deviations of the software from the system specification when receiving client invocations or serving requests from external smart contracts.

To overcome the issues mentioned above, let us note that a smart contract can interact with external data sources by using the so-called oracle service [48], which is a trusted intermediary that helps smart contracts to access and fetch external data not stored on the blockchain. Notwithstanding, oracle services represent a valuable mechanism to improve the proper coding of blockchain-based applications logic; oracle services also represent a weakness as they can be compromised, e.g., providing maliciously tampered data or becoming unresponsive. The solution within the current literature is to compare data received from various oracles to detect any misbehavior or associate a proof-of-correctness to the returned data. However, these solutions may have performance inefficiencies, and thus we present a different solution of implementing the oracle service within the context of Trusted Execution Environment (TEE) [49].

A TEE solution combines hardware and software mechanisms to have the system execution context divided into two execution environments. The first one is named the Rich Execution Environment (REE) and includes the standard Operative System (OS) with extensive features and a significantly wide attack surface. The second one is the proper TEE and hosts a Secure OS offering libraries to invoke sensitive operations, such as cryptographic primitives. It is used to support secure usage of the key elements of the machine, e.g., a secure communication channel between the processor and the I/O peripherals, memory isolation between common and sensitive data, and the hosting of user software with critical functions. In our study, we used a software-based TEE by having the machine being characterized by the Monitor mode in addition to the REE and TEE modes to help the context switch from REE to TEE by marshaling the transition request and mediating the interaction of the user application with the isolated execution environment. Such a solution limits the possibility of having the oracle behavior being compromised by an attacker. To protect the interactions over the insecure network and interact with the oracle in the secure zone, we have used the Transport Layer Security (TLS) to let the software isolated in the TEE of the nodes hosting one ledger one hosting the other one. Furthermore, the oracles authenticate themselves by using a certificate mechanism.

Figure 5 schematically illustrates the proposed solution. A gateway is a software application receiving requests from the user, and based on them, invokes a given smart contract deployed on the hosted blockchain platform. Suppose the invoked smart contract imposes a transfer of an asset towards another blockchain. In that case, the smart contract interacts with the oracle service hosted in the secure zone of the TEE, which establishes a TLS connection with its counterpart at the companion node. The latter receives the request, checks the current state of the blockchain, and performs an update if the request action is doable considering the fetched blockchain state. Finally, the outcome is returned to the oracle, which replies to the smart contract in charge of appropriately updating the first blockchain state.

The user invocation of the functions provided by the gateway is another weakness to be secured properly by implementing authentication and authorization to avoid interactions with malicious adversaries. Permissioned blockchains use a certificate-based approach for the access control of the users, but permissionless ones do not provide them as they allow anyone to interact with the infrastructure. To also protect permissionless solutions, we propose a TEE-based scheme where the code running in the TEE represents the Root of Trust (RoT) as it is tamper-resistant, thus avoiding adopting a complex certification-based architecture.

Before inserting any transaction, a user should contact the TEE to obtain a password, as, without a password, it is not possible to perform any transaction on the distributed ledger. To this purpose, as Figure 6 shows, the gateway forwards the received command to the local TEE through a protected and ciphered channel established with TLS. The latter generates a random password and sends it to the user’s email as an encrypted authentication token (similarly to what happens in the JSON Web Tokens [50]) through a third-party HTTPS web service. With the received password, the user can perform transactions on the ledger or even between two or more ledgers by invoking smart contracts and using the gateway. In addition, the gateway forwards the received password to the local TEE, which compares it with the previously generated one and reports back the authentication decision, as Figure 7 illustrates.

## 4. Implementation and Experimental Results

This section presents the prototype that implements the proposed approach presented in the previous section. Such an implementation is based on two different blockchain platforms, i.e., Hyperledger Fabric and Sawtooth, plus the TEE solution named Open Enclave. The first three subsections briefly introduce them, while the last one specifies how they have been used for prototyping the solution. Finally, we present some experimental results that we collected with the implemented prototype that demonstrate the feasibility of the proposed solution.

### 4.1. Hyperledger Fabric

A notable characteristic of Hyperledger Fabric is it adopts an optimistic model with the primary objective of increasing the efficiency of transaction generation, with the drawback of potentially invalidating some of them. Its model is based on the execute-order-validate architecture rather than the traditional order-execute. In particular, the Hyperledger Fabric executes smart contracts at the very first step. It orders concurrently running transactions, and finally, it verifies their validity, i.e., it applies new transactions on ledgers after verifying that their requirements are still fulfilled. On the contrary, the traditional order-execute approach first orders concurrent transactions and executes related smart contracts in a deterministic manner. Consequently, Hyperledger Fabric transactions can be efficiently validated in parallel, as the ordering phase runs after the execute one. However, some transactions already completed may fail the validation phase, e.g., as they try to sell the same unique item twice (double-spending). In this case, transactions are added to the ledger anyway but flagged as invalid.

In Hyperledger Fabric, nodes belong to different organizations, and for each organization, there is a Membership Service Provider (MSP) providing them with credentials and identities (embedded in an X.509 certificate). Nodes can have the following primary roles:Clients: This requires creating a new transaction based on a specific endorsement policy, detailing how to select nodes involved in a transaction creation procedure. To this purpose, clients (i) contact a subset of endorser peers as specified by the endorsement policy, e.g., at least one for each organization involved in the transaction; (ii) wait for a given amount of transaction endorsements, again as specified by the endorsement policy, e.g., majority or all; and (iii) finally send the new transaction to orderers.Peers: multi-role nodes executing smart contracts, validating transactions provided by clients, and maintaining a local copy of the ledger by committing transactions;Committers, nodes with the only role of maintaining the ledger and updating it whenever they receive a new block.Endorsers: nodes actually executing a smart contract whenever they receive a proposal of transaction. During the endorsement of a new transaction, endorsers securely sign so-called endorsement messages (also containing transaction output, transaction id, endorser id, and endorser signature) and send them to the client requiring the new transaction.Orderers: nodes are collecting requests of new transactions creations, grouping multiple transactions in a block, e.g., sorting concurrent transaction requests coming from different clients and issuing commands to peers to add new blocks on top of the ledger. Note that orderers are unaware of transaction semantics and exploit cryptographic signatures of endorsers to create new blocks.

Smart contracts related to the same application are typically grouped in a unique Hyperledger Fabric Chaincode. Each smart contract allows to specify how a new transaction can be created, also detailing inputs and outputs. Once endorsers have successfully run a smart contract, it is possible to create a new transaction, but it is the orderer in charge of adding the transaction to a new block.

### 4.2. Hyperledger Sawtooth

Sawtooth is a highly modular open-source project for building enterprise permissioned (private) blockchains by separating the core system from the applications without knowing the underlying structure of the core system. Smart contracts can be implemented by using various possible languages, such as Java, Go, Rust, Python, JavaScript, and Swift, each with its own transaction processors that the users can customize. Moreover, it leverages the Seth transaction family to be compatible with Ethereum Contract written in Solidity. On the contrary to Fabric that requires serial transaction execution, Sawtooth is equipped with an advanced parallel scheduler to allow parallel execution of transactions. Considering the data in the blockchain needed for the transactions to be executed, Sawtooth isolates them while keeping valid contextual changes. Moreover, Sawtooth defines a proper eventing service within its core elements to decouple the various parts composing its internals. This allows applications to subscribe to events that affect the blockchain, apply specific events defined by a transaction family, and relay information about the validity of a transaction to the clients without storing that data in the state.

As Fabric, Sawtooth supports multiple consensus algorithms that the administrator can easily select at deployment time. The consensus is chosen and defined during the initial configuration of the blockchain. Still, it is also possible to change the consensus mechanism after the blockchain has been created through one or two transactions. In addition, it also provides the possibility for developers to implement their own consensus rules so that the platform can be easily extended with novel algorithms.

### 4.3. Open Enclave

Open Enclave is an SDK for building hardware-agnostic TEE-based applications in C and C++, where the overall application is partitioned in an untrusted Component or REE, called the host, and a trusted Component or TEE, called the enclave. An enclave is a secure container whose memory (code and data) is protected from access by outside entities or other enclaves to offer confidentiality for data and code execution. Isolation is provided by hardware mechanisms, such as Intel Software Guard Extensions (SGX) that are sets of instructions to protect from disclosure or modification. However, the SDK has been designed to generalize the development of enclave applications across different hardware vendors, such as OP-TEE OS on ARM TrustZone.

A host is an application running in the normal user mode, creating and interacting with an enclave, hosting small sensitive functions. The code in the enclave needs to be a compact and small piece of code to minimize the Trusted Computing Base, i.e., the set of tasks that are critical to system security. The SDK defines call-ins and call-outs mechanisms for the communication with the enclave, ad depicted in Figure 8, and the data marshaling schemes associated with them. Specifically, the host invokes a function in the enclave by using the so-called ECALL. The code in the enclave can invoke a function in the host, such as a system call, by using the OCALLs. The parameters for ECALL/OCALL invocations are appropriately sanitized and marshaled to secure the interaction from possible software vulnerabilities.

### 4.4. Prototype Realization

The implemented proof of concept is based on an Hyperledger Sawtooth application, called the SimpleWallet system [51], acting as the entry point to allow customers to deposit/withdraw/transfer money. The SimpleWallet application is composed of two different components written in Python: the processor and the client. The processor receives a payload from the client and extracts from it the parameters of the requested operation, such as the operation command, the amount involved in the operation, and the key address of the wallet involved in the operation, and in case of a transfer also the destination wallet. In the case of a money transfer, if the balance in the origin wallet is enough for the transfer, the transfer can take place. If Sawtooth does not manage the destination wallet, the function calls the appropriate method for sending the tokens to a BasicTransferAsset account. Otherwise, it is an internal transfer operation that involves two accounts that belong to the SimpleWallet domain.

The operation of transferring to Fabric first constructs the command sent to the enclave, that will encrypt and relay the command to Fabric domain. Then, in the reverse order, a response is sent from Fabric to SimpleWallet, indicating the successful execution of the request. If a positive response is received, the balance of the sender’s wallet will be uploaded by withdrawing the sent amount. Otherwise, the user will be notified of “Transaction Refused”. The communication with the enclave is done by establishing a channel using an SSL socket with the host on the address 192.168.1.199 on port 8082. The server uses a certificate, where the hostname is set as “NAME”. When the sending is done, it waits for a response of 1024 bytes. When the response is received, it closes the connection and returns the answer to the caller method.

The enclave-sawtooth and enclave-fabric software implementations are similar to each other; there are very few differences between them and both are available at https://github.com/chrespo-prof/TEEBlockchainInteroperability.git (accessed on 2 July 2021). Both host and enclave scripts are written in the C programming language, using the OpenEnclave SDK, crypto, and OpenSSL libraries. In particular, the OpenSSL library is useful for providing the SSL connection on the socket. The Sawtooth host is responsible for establishing a server socket to receive messages from SimpleWallet through an SSL connection. When it receives a message, it sends the command to the enclave that encrypts the message. The encrypted message is then sent back to the host that relays the message to the Fabric host. Intuitively, the Fabric host is similar but waits for messages from the Sawtooth host. The Sawtooth enclave encrypts the received message with its private key and initial vector. The only one that could decrypt the message is the Fabric enclave. When the Fabric host receives the message, it passes it to the enclave to decrypt it. The enclave returns the decrypted message to the host. This last one will relay the decrypted message to the BasicTransferAsset to perform the asset transfer from one system to another. Both hosts will wait for the responses that should be provided in inverse order. Therefore, once the BasicTransferAsset has received and analyzed the message, it returns the positive or negative response back to the Fabric host. This response will be sent back until it reaches the SimpleWallet system again, bypassing the Sawtooth host again. The encryption function of the enclave is based on the use of “mbedtls” library for using the encryption AES-CBC-256 method. In this function, two particular parameters cannot be stolen by outside because this function is executed in a protected environment. This method first does some operations to construct the formatting of the data; then, it inserts in the variables the information necessary to perform the encryption.

For user authentication, another enclave application has been implemented. The application can assign a randomly generated password and pass it to the user e-mail account or check the message correspondence with its own internal password variable to grant action permission. To send an e-mail, an internal ad hoc library is used for establishing an in-enclave secure communication with a web server. Specifically, we have used a server written in Python that uses some API exposed from the service If This Then do That (IFTTT) [52], a free web service that allows the creation of simple chains of conditions, called applets. In our case, the triggering action was represented by an HTTPS get the call to an URL, and the web service was the e-mail forwarding.

### 4.5. Experimental Results

We have tested the execution of interoperable operations between Sawtooth and Fabric, both running in two Virtual Machines using Virtual Box hosted within a MacBook Pro laptop equipped with Intel Core i7 6-core at 2.6GHz with a 1TB SSD memory, whose results are illustrated in Figure 9. First, we request to transfer assets in the BasicTransferAsset system by indicating a destination walled in Fabric, as represented in Figure 9a. Next, the request goes through our software, and a message is sent towards the host on the node running a Fabric node that passes it to the enclave and performs the transfer, as depicted in Figure 9b. Then, the message is sent back to the process on the node with the Sawtooth, and the origin wallet is updated, as verified in Figure 9c.

We have measured the latency to operate blockchains and compared it with the same operation between wallets on the identical blockchain, shown in Figure 10. The testbed is composed of two virtual machines on one computer hosting elements for a test network on Sawtooth. Another computer hosts two virtual machines where all the nodes for a Fabric test network are deployed. In Figure 10a, we can see that to perform the transaction between a blockchain and another, the proposed solution takes about 3 s and 50 ms, while a local transfer on Sawtooth takes 13 ms. Thus, despite being a considerable increase, 3 s for a transaction in the considered use case is tolerable. Moreover, in Figure 10b, we see that the introduction of the TEE-verification when making an operation to get the balance of a wallet implies an additional latency, but of a minimal amount, i.e., no more than 3 ms on average.

## 5. Conclusions

Industry 4.0 and other novel ICT visions as Smart Cities or personalized and remote healthcare are mainly composed as an ecosystem of ICT infrastructures and companies rather than built from scratch by a single organization. It is not possible to impose a single technology solution on the ecosystem participants in such a context. Moreover, while the blockchain is seen as an enabling technology to support the advent of Industry 4.0, it is supported by many different solutions, which are not interoperable. Therefore, the research community has recently started to investigate the problem of blockchain interoperability and to propose various kinds of solutions, which have been unable to resolve the problem fully. In this work, we have presented a relay scheme empowered by the TEE so as to offer a more secure and efficient interoperability solution. We have proved that such a solution can let two different blockchains transfer assets and interoperate. At the same time, the consequent performance worsening is acceptable within the context of the considered application scenario. These results indicate that, as usual, security is achieved at higher performance costs and reduced scalability, so further research is still needed to achieve the optimal trade-off among these opposing demands. This is challenging if blockchain interoperability has to be achieved within applications where guaranteeing lower and predictable latency is as critical as preventing possible attacks.

As a future plan, we aim to study how to let multiple blockchains be seamlessly interoperable by simplifying the joining of a new blockchain to the federation and multicasting the cross-chain requests among the federation participants. We have integrated the blockchain into the TEE solution by an ad hoc solution. Still, for this aim, it is also possible to use Hyperledger Avalon [14], which supports only off-chain computation with Fabric, Ganache [53] that is a personal Ethereum blockchain, and Hyperledger Besu [54] that is an Ethereum client. We have done preliminary benchmarking with Avalon on Fabric and compared it with our solution, and the performance is comparable. Therefore, our plan for future work is also to extend our solution by using Avalon and implement some more connectors for other unsupported blockchain platforms to obtain a standard-based solution, as Avalon implements the Trusted Compute Specifications published by the Enterprise Ethereum Alliance [55]. We also aim at comparing the performance of the proposed solution with the one achievable with the main competitors listed in Table 1, which has not been possible in this work as they still lack connectors to the considered blockchain solutions.

## Figures and Tables

**Figure 1 sensors-21-04955-f001:**
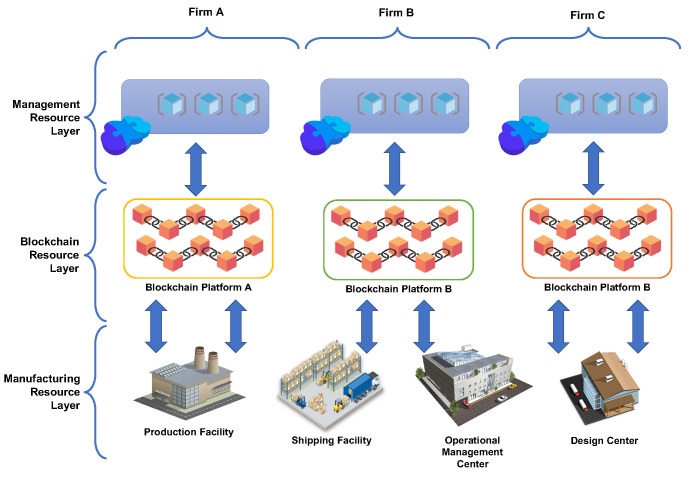
Blockchain integration within a generic industry process for collaborative manufacturing.

**Figure 2 sensors-21-04955-f002:**
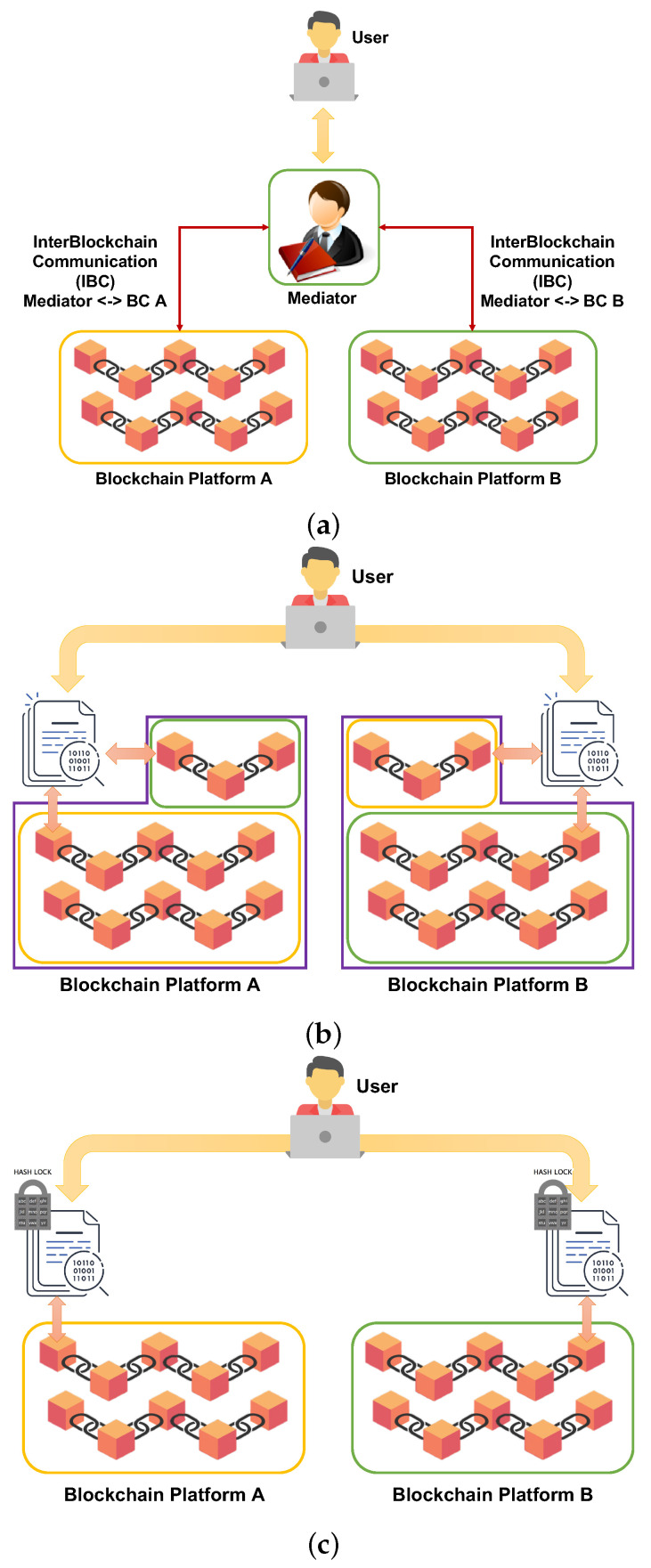
Blockchain interoperability schemes. (**a**) Notary scheme solution. (**b**) Sidechain-based approach. (**c**) Hash-locking smart contracts for asset transfer between blockchains.

**Figure 3 sensors-21-04955-f003:**
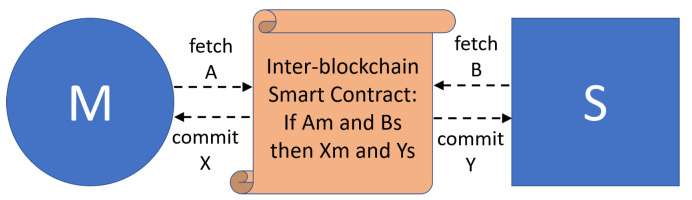
Inter-blockchain smart contract.

**Figure 4 sensors-21-04955-f004:**
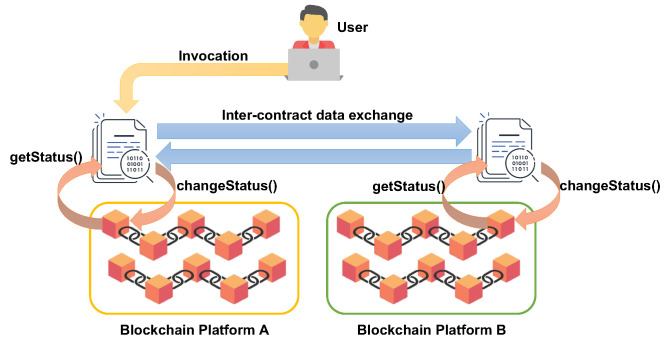
Inter-blockchain smart contract implementation approach.

**Figure 5 sensors-21-04955-f005:**
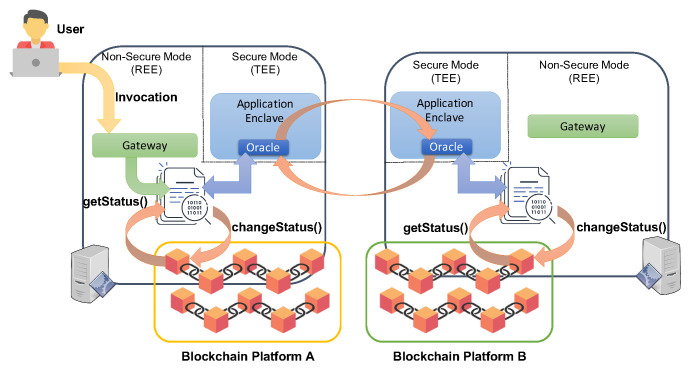
Proposed TEE-based interoperability solution.

**Figure 6 sensors-21-04955-f006:**
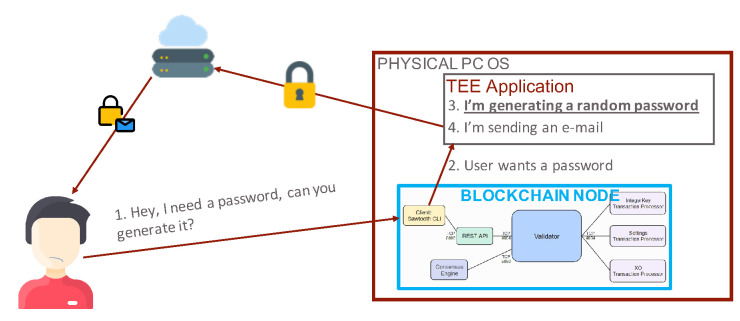
Password generation phase.

**Figure 7 sensors-21-04955-f007:**
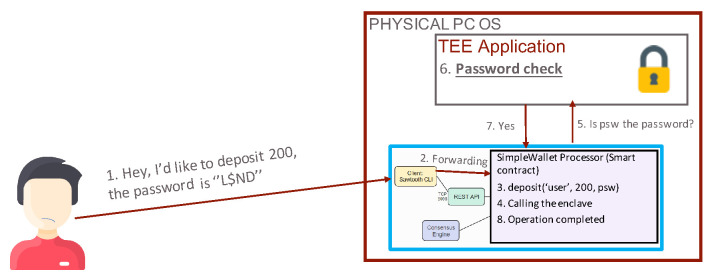
Password verification phase.

**Figure 8 sensors-21-04955-f008:**
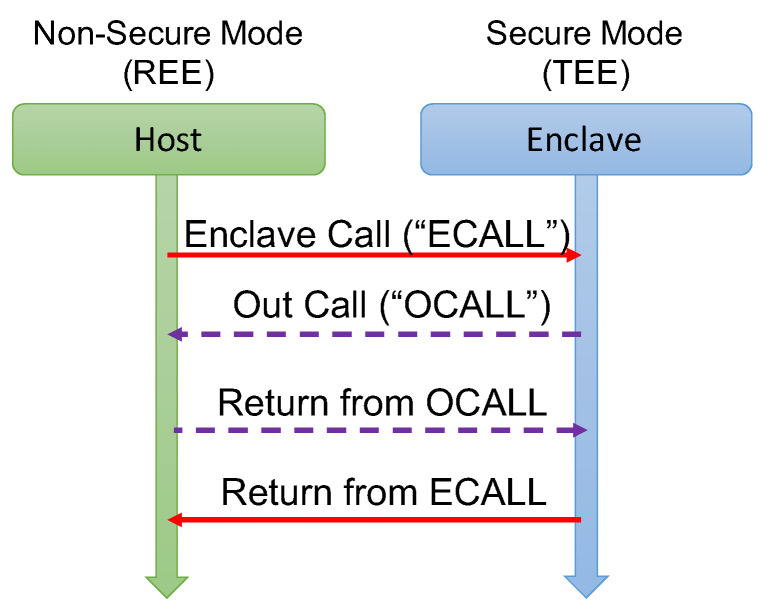
Interaction between Host and Enclave by using ECall/OCall.

**Figure 9 sensors-21-04955-f009:**
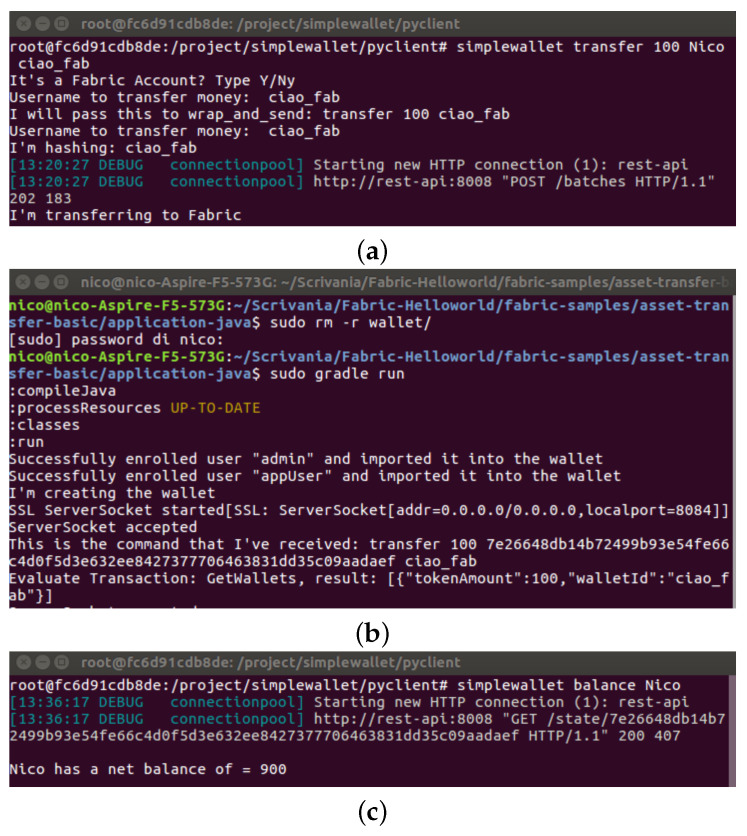
Interoperability unit test: (**a**) Transferring from a Sawtooth wallet to a Fabric one, (**b**) Receiving the transfer request on Fabric, (**c**) Checking the origin wallet after the transfer.

**Figure 10 sensors-21-04955-f010:**
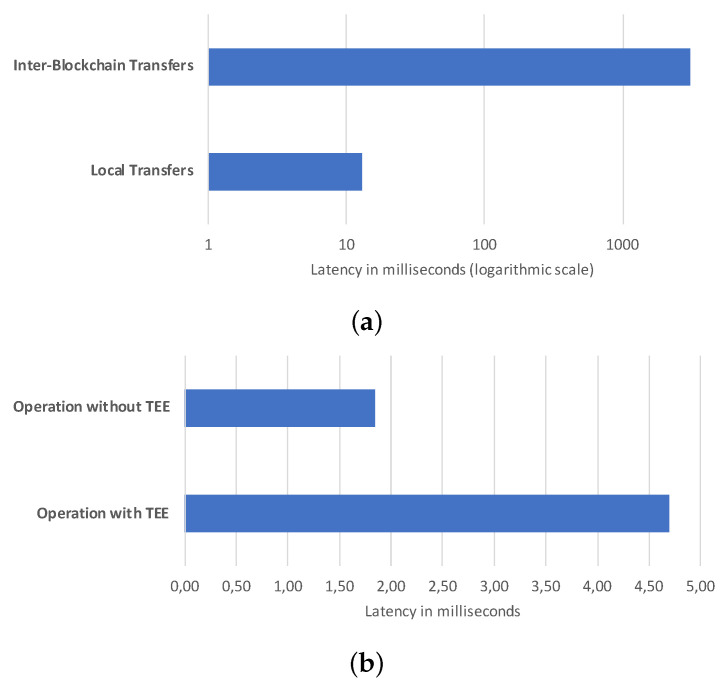
Latency assessment: (**a**) Latency worsening when transferring among blockchains, and (**b**) Latency worsening when using TEE-based authentication.

**Table 1 sensors-21-04955-t001:** Analysis of solutions to have interoperable blockchains.

Solution	Approach	Security Mechanisms	Cons
Hyperledger Cactus [41]	Notary Scheme	Details Missing	Centralized Cactus Node Server, which can be compromised	
Polkadot [42]	Sidechain + Relay Nodes	Shared state between the relay chain and connected parachains	Validators can be compromised, use of costly BFT consensus among blockchains	
Cosmos [43]	Sidechain + Relay Nodes	Tendermint Core BFT consensus mechanism and 100-validator node network maintain security	Validators can be compromised, use of costly BFT consensus among blockchains and the deployment of a centralized hub	
ChainLink [44]	Network of Relay Nodes	TEE-based connector between on-chain and off-chain systems, with secured adapters towards various blockchains	ERC-20 tokens provide data to any connected blockchain, with no support to shared state	
Interledger [45]	Network of Relay Nodes	Conditional transfers to secure payments	Interledger connectors can be compromised	
Lightning Network [46]	Hash-locking scheme	Multiple multi-signature channels established among heterogeneous blockchains	Validators can be compromised, only for micropayments	

## Data Availability

Data is contained within the article.

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
