# Peer review of "Interoperable Blockchains for Highly-Integrated Supply Chains in Collaborative Manufacturing"

_sensors, 2021, doi:10.3390/s21154955_

Round 1
Reviewer 1 Report
Dear authors,
I thank you for having trusted Sensors for your work.
In my opinion your work is very valuable, and has a lot of interest and potential. However I think you should try to redo it in such a way that the following conditions are met:
1. the experiments are replicable. For this it would be necessary to know every detail, both the software and the hardware used. It is not possible at this time to adequately evaluate their work without having the code. It is not enough to show the results as you have done in section 4.
2. the results are quantitatively compared with other previous models and the performance of the new model is demonstrably better than the previous ones in quality, performance, security, or whatever dimension you decide to highlight.
Other minor aspects that could be highlighted are the following:
1. the paper would benefit if you would present a "graphical Abstract" in the Introduction. this would help the reader.
2. in my opinion, you have overused footnotes. You should reduce them to a minimum and use proper citation.
3. in the introduction the paragraph between lines 74 and 94 is of vital importance. However, it is not accessible to the general public. It should be re--written to ensure that it is understandable.
4. in line 131 you repeat "only".
5. The Figures should be retouched to ensure proper interpretation. Many figures provide no information or are illegible.
Thank you very much!
Author Response
Q1 The experiments are replicable. For this it would be necessary to know every detail, both the software and the hardware used.
R1: We agree with the reviewer and for this reason we clearly indicated the used hardware and uploaded our implementation on GitHub and stated its link within the manuscript.
Q2 The results are quantitatively compared with other previous models and the performance of the new model is demonstrably better than the previous ones.
R2: We have inserted a qualitative comparison of the existing solutions in Section 2. In the conclusions, we have indicated as a possible future work a comparison with existing solutions.
Q3 The paper would benefit if you would present a ”graphical Abstract” in the introduction.
R3: We thank the reviewer for pointing out this aspect, we have created an image as graphical abstract and uploaded it in this revision.
Q4 You should reduce the footnotes and use proper citation.
R4: We followed the reviewer suggestion and included the cited web sites in the reference list.
Q5 In the introduction the paragraph between lines 74 and 94 is of vital importance. However, it is not acceptable to the general public. It should be re-written to ensure that it is understandable.
R5: We followed the reviewer suggestion and improved the presentation by inserting proper clarifications of the used terms and concepts.
Q6 The figures should be retouched to ensure proper interpretation.
R6: We have corrected Figures 9 and 10 where some text was not readable as covered by the following subfigure.
Q7 In line 131 you repeat ”only”.
R7: We have corrected this, and also other language errors.
Finally, we would like to thank Reviewer 1 for your considerable comments and advice, which helps us revise the manuscript.

Reviewer 2 Report
Interested topic and interesting study but some imperfection must be improved:
- add clear aim, methodolgy and study results in the abstract. now it is not enough informative
- Authors proposed new solution but forgot to provice comprehansive study on the literature review - I cannot see also clear research questions or hypothesis after literature review as well as results dicussion is very limited
- Add practical recommendations from this study
Author Response
Q1 Add clear aim, methodology and study results in the abstract
R1: We have followed the received comment and added new statements containing the aim, methodology and results in the abstract.
Q2 Authors forgot to provide comprehensive study on the literature review. I cannot see also clear research questions or hypothesis after literature review as well as results discussion is very limited.
R2: We thank the reviewer to pointing this out and we have improved our manuscript by inserting two research questions at the end of Section 2, and result discussion in the abstract. We have enlarged the analysis of the literature with a table of comparison of the existing solutions. Finally, we have added
in in-depth analysis review on Blockchain and Industry 4.0 in Section 2.2.
Q3 Add practical recommendations from this study.
R3: We thank the reviewer for highlighting the missing element in the manuscript. We have introduced them within the final section of the paper.
Finally, we would like to thank again Reviewer 2 for your considerable comments and advice, which helps us revise the manuscript.

Round 2
Reviewer 1 Report
Thank you for the changes in the manuscript. Work can be published as is.
Best regards.